# ALIGNING BRAIN FUNCTIONS BOOSTS THE DECODING OF VIDEOS IN NOVEL SUBJECTS

## ABSTRACT

Deep learning is leading to major advances in the realm of brain decoding from functional Magnetic Resonance Imaging (fMRI). However, the large inter-subject variability in brain characteristics has limited most studies to train models on one subject at a time. Consequently, this approach hampers the training of deep learning models, which typically requires very large datasets. Here, we propose to boost brain decoding by aligning brain responses to videos across subjects. Compared to the anatomically-aligned baseline, our method improves out-of-subject decoding performance by up to 75%. Moreover, it also outperforms classical single-subject approaches when fewer than 100 minutes of data is available for the tested subject. Furthermore, we propose a new multi-subject alignment method, which obtains comparable results to that of classical single-subject approaches while easing out-of-subject generalization. Finally, we show that this method aligns neural representations in accordance with brain anatomy. Overall, this study lays foundations to leverage extensive neuroimaging datasets and enhance the decoding of individuals with a limited amount of brain recordings.

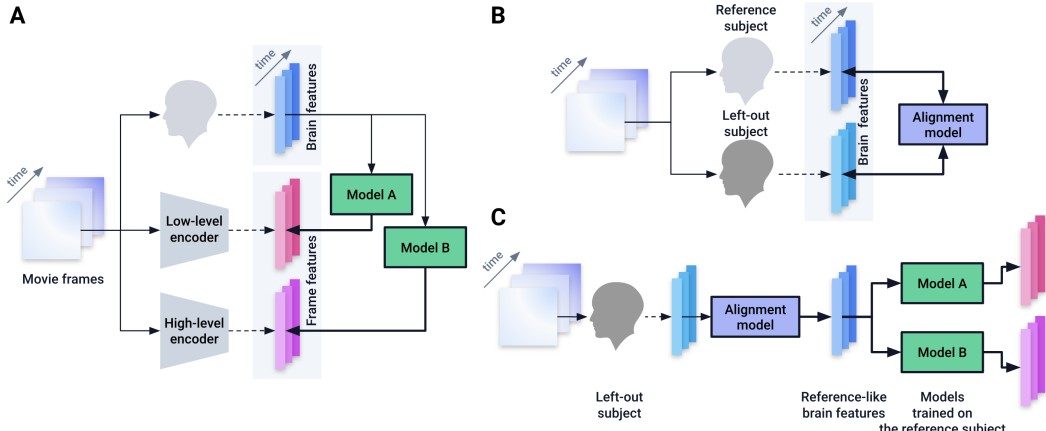

Figure 1: **General outline of video decoding from BOLD fMRI signal in left-out subjects**
**A.** For every image associated with a brain volume, one computes its low-level and high-level latent representations using pre-trained models. Subsequently, regression models can be fitted to map brain features onto each of these latent representations. **B.** BOLD signal acquired in two subjects watching the same movie can be used to derive an alignment model which associates voxels of the two subjects based on functional similarity. **C.** Once this alignment model is trained, it can be used to transform brain features of the left-out subject into brain features that resemble that of the reference subject. In particular, this allows one to use models that have been trained on a lot of data coming from a reference subject data, and apply it on a left-out subject for whom less data was collected.

# 1 INTRODUCTION

**Decoding the brain** Deep learning is greatly accelerating the possibility of decoding mental representations from brain activity. Originally restricted to linear models (Mitchell et al., 2004; Harrison & Tong, 2009; Haynes & Rees, 2006), the decoding of brain activity can now be carried out with deep learning techniques. In particular, using functional Magnetic Resonance Imaging (fMRI) signals, significant progress has been made in the decoding of images (Ozcelik & VanRullen, 2023; Chen et al., 2023a; Scotti et al., 2023; Takagi & Nishimoto, 2023; Gu et al., 2023; Ferrante et al., 2023; Mai & Zhang, 2023), speech (Tang et al., 2023), and videos (Kupershmidt et al., 2022; Wen et al., 2018; Wang et al., 2022; Chen et al., 2023b; Lahner et al., 2023; Phillips et al., 2022).

**Challenge** However, brain representations are highly variable across subjects, which makes it challenging to train the same model on multiple subjects. Therefore, with few noteworthy exceptions (Haxby et al., 2020; Ho et al., 2023), studies typically train a decoder on a single subject at a time. With this constraint in mind, major effort has been put towards building fMRI datasets collecting a lot of data in a limited number of participants (Allen et al., 2022; Wen et al., 2017; LeBel et al., 2023; Pinho et al., 2018). Nonetheless, the necessity to train and test models on a single subject constitutes a major impediment to using notoriously data-hungry deep learning approaches.

**Functional alignment** Several methods can align the functions – as opposed to the anatomy – of multiple brains, and thus offer a potential solution to inter-subject variability: differentiable wrappings of the cortical surface (Robinson et al., 2014), rotations between brain voxels in the functional space (Haxby et al., 2011), shared response models (Chen et al., 2015; Richard et al., 2020), permutations of voxels minimizing an optimal transport cost (Bazeille et al., 2019), or combinations of these approaches (Feilong et al., 2022). However, it is not clear which of these methods offers the best performance and generalization capabilities (Bazeille et al., 2021). Besides, several studies rely on deep learning models trained in a self-supervised fashion to obtain a useful embedding of brain activity, in hope that this embedding could be meaningful across subjects (Thomas et al., 2022; Chen et al., 2023a). However, it is currently unknown whether any of these methods improve the decoding of naturalistic stimuli such as videos, and how such hypothetical gain would vary with the amount of fMRI recording available in a given a subject.

**Approach** To address this issue, we leverage fMRI recordings of multiple subjects to boost the decoding of videos in a single left-out subject. This requires fitting two models: an alignment model and a decoder. The alignment aims at making brain responses of a left-out subject most similar to those of a reference subject. Here, we leverage optimal transport to compute this transformation using functional and anatomical data from both subjects. The decoder consists of a linear regression trained to predict the latent representations of movie frames from the corresponding BOLD signals.

We evaluate video decoding in different setups. In particular, we assess (1) whether training a decoder with several subjects improves performance, (2) whether decoders generalize to subjects on which they were not trained and (3) the extent to which functional alignment improves aforementioned setups.

**Contributions** We first confirm the feasibility of decoding, from 3T fMRI, the semantics of videos watched by the subjects. Our study further makes three novel contributions:

1. functional alignment across subjects boosts video decoding performance when left-out subjects have a limited amount of data
2. training a decoder on multiple aligned subjects reaches the same performance as training a single model per subject
3. the resulting alignments, computed from movie watching data, yield anatomically-coherent maps.

From a representation learning perspective, this is one more piece of evidence that representations learnt by deep learning models can help model and decode brain signal, even with stimuli as complex as naturalistic videos. Our results also show that, in high-data regimes, naturalistic movie-watching yields functional features which can help discriminate between parts of the cortex much beyond the visual system.

## 2 METHODS

Our goal is to decode visual stimuli seen by subjects from their brain activity. To this end, we train a linear model to predict latent representations – shortened as *latents* – of these visual stimuli from BOLD fMRI signals recorded in subjects watching naturalistic videos.

In the considered data, brains are typically imaged at a rate of one scan every 2 seconds. During this period, a subject sees 60 video frames on average. For simplicity, we consider the restricted issue of decoding only the first video frame seen by subjects at each brain scan. Formally, for a given subject, let $\boldsymbol{X} \in \mathbb{R}^{n,v}$ be the BOLD response collected in $v$ voxels over $n$ brain scans and $\boldsymbol{Y} \in \mathbb{R}^{n,m}$ the $m$-dimensional latent representation of each selected video frame for all $n$ brain scans.

### 2.1 BRAIN ALIGNMENT

**Anatomical alignment**  As a baseline, we consider the alignment method implemented in Freesurfer (Fischl, 2012), which relies on anatomical information to project each subject onto a surface template of the cortex (in our case *fsaverage5*). Consequently, brain data from all subjects lie on a mesh of size $v = 10\,242$ vertices per hemisphere.

**Functional alignment**  On top of the aforementioned anatomical alignment, we apply a recent method from Thual et al. (2022) denoted as Fused Unbalanced Gromov-Wasserstein (FUGW) [1]. As illustrated in Figure 1.B, this method consists in using functional data to train an alignment that transforms brain responses of a given left-out subject into the brain responses of a reference subject. This approach can be seen as a soft permutation of voxels [2] of the left-out subject which maximizes the functional similarity to voxels of the reference subject.

Formally, for a left-out subject, let $\boldsymbol{D}^{\text{out}} \in \mathbb{R}^{v,v}$ be the matrix of anatomical distances between vertices on the cortex, and $\boldsymbol{w}^{\text{out}} \in \mathbb{R}^v_+$ a probability distribution on vertices. $\boldsymbol{w}^{\text{out}}$ can be interpreted as the relative importance of vertices ; without prior knowledge, we use the uniform distribution. Reciprocally, we define $\boldsymbol{D}^{\text{ref}}$ and $\boldsymbol{w}^{\text{ref}}$ for a reference subject. Note that, in the general case, $v$ can be different from one subject to the other, although we simplify notations here.

We derive a transport plan $\boldsymbol{P} \in \mathbb{R}^{v,v}$ to match the vertices of the two subjects based on functional similarity, while preserving anatomical organisation. For this, we simultaneously optimize multiple constraints, formulated in the loss function $\mathcal{L}(\boldsymbol{P})$ described in Equation 1:

$$
\mathcal{L}(\boldsymbol{P}) \triangleq (1-\alpha) \underbrace{\sum_{0 \le i,j < n} ||\boldsymbol{X}^{\text{out}}_i - \boldsymbol{X}^{\text{ref}}_j||^2_2 \boldsymbol{P}_{i,j}}_{\text{Wasserstein loss}} + \alpha \underbrace{\sum_{0 \le i,k,j,l < n} |\boldsymbol{D}^{\text{out}}_{i,k} - \boldsymbol{D}^{\text{ref}}_{j,l}|^2 \boldsymbol{P}_{i,j}\boldsymbol{P}_{k,l}}_{\text{Gromov-Wasserstein loss}}
$$

$$
+ \rho \Big( \underbrace{\text{KL}(\boldsymbol{P}_{\#1} \otimes \boldsymbol{P}_{\#1} \,|\, \boldsymbol{w}^{\text{out}} \otimes \boldsymbol{w}^{\text{out}}) + \text{KL}(\boldsymbol{P}_{\#2} \otimes \boldsymbol{P}_{\#2} \,|\, \boldsymbol{w}^{\text{ref}} \otimes \boldsymbol{w}^{\text{ref}})}_{\text{Marginal constraints}} \Big) + \varepsilon \underbrace{\text{H}(\boldsymbol{P})}_{\text{Entropy}} \quad (1)
$$

with $\boldsymbol{P}_{\#1} \triangleq (\sum_j \boldsymbol{P}_{i,j})_{0 \le i < v}$ and $\boldsymbol{P}_{\#2} \triangleq (\sum_i \boldsymbol{P}_{i,j})_{0 \le j < v}$ the first and second marginal distributions of $\boldsymbol{P}$, $\otimes$ the Kronecker product between two matrices, and $\text{KL}(\cdot, \cdot)$ the Kullback-Leibler divergence. $\alpha$, $\rho$ and $\varepsilon$ are hyper-parameters setting the relative importance of each constraint.

Following Thual et al. (2022), we minimize $\mathcal{L}(\boldsymbol{P})$ with 10 iterations of a block coordinate descent algorithm (Séjourné et al., 2021), each running $1\,000$ Sinkhorn iterations (Cuturi, 2013). Subsequently, we define $\phi_{\text{out}\to\text{ref}} \colon \boldsymbol{X} \mapsto \big(\boldsymbol{P}^T \boldsymbol{X}^T\big) \oslash \boldsymbol{P}_{\#2} \in \mathbb{R}^{n,v}$ where $\oslash$ is the element-wise division, a function which transports any matrix of brain features from the left-out subject to the reference subject. To simplify notations, for any $\boldsymbol{X}$ defined on the left-out subject, we define $\boldsymbol{X}^{\text{out}\to\text{ref}} \triangleq \phi_{\text{out}\to\text{ref}}(\boldsymbol{X})$.

---

[1] https://alexisthual.github.io/fugw

[2] We use the words *voxel* (volumetric pixel) or *vertex* (point on a mesh) indifferently.

## 2.2 DECODING

**Brain input**  There is a time *lag* between the moment a stimulus is played and the moment it elicits a maximal BOLD response in the brain (Glover, 1999). Moreover, since the effect induced by this stimulus might span over multiple consecutive brain volumes, we set a *window size* describing the number of brain volumes to aggregate together. To account for these effects, we use a standard Finite Impulse Response (FIR) approach. FIR consists in fitting the decoder on a time-lagged, multi-volume version of the BOLD response. Different *aggregation functions* can be used, such as stacking or averaging. Figure S2 describes these concepts visually.

**Video output**  The matrix of latent features $\boldsymbol{Y}$ is obtained by using a pre-trained image encoder on each video frame and concatenating all obtained vectors in $\boldsymbol{Y}$. Similarly to Ozcelik & VanRullen (2023), and as illustrated in Figure 1.A, we seek to predict CLIP $257 \times 768$ (high-level) and VD-VAE (low-level) latent representations . We use visual – as opposed to textual – CLIP representations (Radford et al., 2021). For comparison, we also reproduce our approach on latent representations from CLIP CLS (high-level) and AutoKL (low-level), which happen to be much smaller [3] and might be computationally easier to fit.

**Model**  Fitting the decoder consists in deriving $\boldsymbol{W} \in \mathbb{R}^{v,m}$, $\boldsymbol{b} \in \mathbb{R}^m$ the solution of a Ridge regression problem – i.e. a linear regression with L2 regularization – predicting $\boldsymbol{Y}$ from $\boldsymbol{X}$.

**Evaluation**  We evaluate the performance of the decoder with retrieval metrics. Let us denote $\boldsymbol{X}$ and $\boldsymbol{Y}$ the brain and latent features used to train the decoder, $\boldsymbol{X}_{\text{test}}$ and $\boldsymbol{Y}_{\text{test}}$ those to test the decoder, and $\hat{\boldsymbol{Y}} \triangleq \boldsymbol{W}\boldsymbol{X}_{\text{test}} + \boldsymbol{b}$ the predicted latents. We ensure that the train and test data are disjoint.

We randomly draw a retrieval set $K$ of 499 frames without replacement from the test data. For each pair $\hat{\boldsymbol{y}}, \boldsymbol{y}$ of predicted and ground truth latents, one derives their cosine similarity score $s(\hat{\boldsymbol{y}}, \boldsymbol{y})$, as well as similarity scores to all latents $\boldsymbol{y}_{\text{neg}}$ of the retrieval set $s(\hat{\boldsymbol{y}}, \boldsymbol{y}_{\text{neg}})$. Let us denote $r(\hat{\boldsymbol{y}}, \boldsymbol{y})$ the rank of $\boldsymbol{y}$, which we define as the number of elements of $K$ whose similarity score to $\hat{\boldsymbol{y}}$ is larger than $s(\hat{\boldsymbol{y}}, \boldsymbol{y})$. In order for the rank to not depend on the size of $K$, we define the *relative rank* as $\frac{r(\hat{\boldsymbol{y}}, \boldsymbol{y})}{|K|}$. Eventually, one derives the median relative rank $\text{MR}(\hat{\boldsymbol{Y}}, K)$:

$$r(\hat{\boldsymbol{y}}, \boldsymbol{y}) \triangleq \left| \left\{ \boldsymbol{y}_{\text{neg}} \in K \mid s(\hat{\boldsymbol{y}}, \boldsymbol{y}_{\text{neg}}) > s(\hat{\boldsymbol{y}}, \boldsymbol{y}) \right\} \right|$$

$$\text{MR}(\hat{\boldsymbol{Y}}, K) \triangleq \text{median}\left( \left\{ \frac{r(\hat{\boldsymbol{y}}, \boldsymbol{y})}{|K|}, \forall (\hat{\boldsymbol{y}}, \boldsymbol{y}) \right\} \right)$$

## 2.3 DECODING AND ALIGNMENT SETUPS

**Within- *vs* out-of-subject**  The *within-subject* setup consists in training a decoder with data $\boldsymbol{X}_{\text{train}}^{S_1}$, $\boldsymbol{Y}_{\text{train}}^{S_1}$ from a given subject, and testing it on left-out data $\boldsymbol{X}_{\text{test}}^{S_1}$, $\boldsymbol{Y}_{\text{test}}^{S_1}$ acquired in the same subject. The *out-of-subject* setup consists in training a decoder with data from a given subject, and testing it on data $\boldsymbol{X}_{\text{test}}^{S_2}$, $\boldsymbol{Y}_{\text{test}}^{S_2}$ acquired in a left-out subject.

**Single- *vs* multi-subject**  The *single-subject* setup consists in training a decoder predicting $\boldsymbol{Y}$ from $\boldsymbol{X}$ for each subject. The *multi-subject* setup consists in training a single decoder using data from multiple subjects. In this case, data from several subjects is stacked together, resulting in a matrix $\boldsymbol{X}_{\text{multi}} \in \mathbb{R}^{n_1 + \dots + n_p, v}$ and $\boldsymbol{Y}_{\text{multi}} \in \mathbb{R}^{n_1 + \dots + n_p, m}$ , where $p$ is the number of subjects.

**Un-aligned *vs* aligned**  In multi-subject and out-of-subject setups, data coming from different subjects can be *aligned* to a *reference* subject. Let us assume that $S_1$ is the reference subject. In the case of multi-subject, all subjects are aligned to $S_1$ and the decoder is trained on a concatenation of $\boldsymbol{X}^{S_1}, \boldsymbol{X}^{S_2 \to S_1}, \dots, \boldsymbol{X}^{S_p \to S_1}$ (see notations introduced at the end of section 2.1) and $\boldsymbol{Y}^{S_1}, \dots, \boldsymbol{Y}^{S_p}$ , where $p$ is the number of subjects. In the case of out-of-subject, it corresponds to aligning $S_2$ onto $S_1$, such that a decoder trained on $S_1$ will be tested on $\boldsymbol{X}_{\text{test}}^{S_2 \to S_1}, \boldsymbol{Y}_{\text{test}}^{S_2}$.

---

[3]Dimensions for CLIP CLS: 768 ; CLIP $257 \times 768 : 257 \times 768 = 197\,376$ ; AutoKL: $4 \times 32 \times 32 = 4\,096$ ; VD-VAE: $2 \times 2^4 + 4 \times 2^8 + 8 \times 2^{10} + 16 \times 2^{12} + 2^{14} = 91\,168$

The aforementioned setups are described visually in Figure 3.A.

**Evaluation under different data regimes**   Note that alignment and decoding models need not be fitted using the same amount of data. In particular, we are interested in evaluating out-of-subject performance in setups where a lot of data is available for a *reference* subject, and little data is available for a *left-out* subject: this would typically be the case in clinical setups where little data is available in patients. In this case, we evaluate whether it is possible to use this small amount of data to align the left-out subject onto the reference subject, and have the left-out subject benefit from a decoder previously trained on a lot of data.

## 2.4 DATASET

We analyze the dataset from Wen et al. (2017). This dataset comprises 3 human subjects who each watched 688 minutes of video in an MRI scanner. The videos consists of 18 train segments of 8 minutes each and 5 test segments of 8 minutes each. Each train segment was presented twice. Each test segment was presented 10 times. Each segment consists of a sequence of roughly 10-second video clips.

The fMRI data was acquired at 3 Tesla (3T), 3.5mm isotropic spatial resolution and 2-second temporal resolution. It was minimally pre-processed with the same pre-processing pipeline than that of the Human Connectome Project (Glasser et al., 2013). In particular, data from each subject are projected onto a common volumetric anatomical template.

Comparably to prior work on this dataset (Wen et al., 2018; Kupershmidt et al., 2022; Wang et al., 2022), we use runs related to the first 18 video segments - 288 minutes - as training data, and runs related to the last 5 video segments as test data.

## 2.5 PREPROCESSING

We implement minimal additional preprocessing steps for each subject separately. For this, we (1) project all volumetric data onto the FreeSurfer average surface template *fsaverage5* (Fischl, 2012), then (2) regress out cosine drifts in each vertex and each run and finally (3) center and scale each vertex time-course in each run. Figure S1 gives a visual explanation as to why the last two steps are needed. The first two steps are implemented with nilearn (Abraham et al., 2014) [4] and the last one with scikit-learn (Pedregosa et al., 2011).

Additionally, for a given subject, we try out two different setups: a first one where runs showing the same video are averaged, and a second one where they are stacked.

## 2.6 HYPER-PARAMETERS SELECTION

To train decoders, we use the same regularization coefficient $\alpha_{\text{ridge}}$ across latent types and choose it by running a cross-validated grid search on folds of the training data. We find that results are robust to using different values and stick to $\alpha_{\text{ridge}} = 50\,000$. Similarly, values for lag, window size and aggregation function are determined through a cross-validated grid search.

Finally, for functional alignment, we stick to default parameters shipped with version 0.1.0 of FUGW. Namely, $\alpha$, which balances between Wasserstein and Gromov-Wasserstein losses – i.e. how important functional data is compared to anatomical data – is set to $0.5$. Empirically, we see that this value yields values for the Wasserstein loss which are bigger than that of the Gromov-Wasserstein loss, meaning that functional data drives these alignments. $\varepsilon$, which controls for entropic regularization – i.e. how blurry computed alignments will be – is set to $10^{-4}$. Empirically, this value yields very anatomically sharp alignments. $\rho$, which sets the importance of marginal constraints – i.e. to what extent more or less mass can be transported to / from each voxel – is set to $1$. Empirically, this value leads to all voxels being transported / matched.

---

[4] https://nilearn.github.io

## 3 RESULTS

### 3.1 WITHIN-SUBJECT PREDICTION OF VISUAL REPRESENTATIONS FROM BOLD SIGNAL AND RETRIEVAL OF VISUAL INPUTS

We report the retrieval predictions of video decoding results in Table 1. For all three subjects of the Wen et al. (2017) dataset, and for all four types of latent representations considered, a Ridge regression fitted within-subject achieves significantly above-chance performance. Besides, performance varies across subjects, although well-performing subjects reach good performance on all types of latents.

Results reported in Table 1 were obtained for a lag of 2 brain volumes (i.e. 4 seconds since TR = 2 seconds) and a window size of 2 brain volumes which were averaged together (see definitions in section 2.5). These parameters were chosen after running a grid search for lag values ranging from 1 to 5, a window size ranging from 1 to 3, and 2 possible aggregation functions for brain volumes belonging to the same window (namely averaging and stacking). Figure S4 shows results using the averaging aggregation function for different values of lag and window size, averaged across subjects. Eventually, these results were obtained by stacking all runs of the training dataset, as opposed to averaging repetitions of the same video clip. The two approaches yielded very similar metrics. We expand on this matter in section 3.3.

Finally, Figure 2 shows retrieved images for Subject 2. Qualitatively, we observe that retrieved images often fit the theme of images shown to subjects (with categories like indoor sports, human faces, animals, etc.), but also regularly exhibit failure cases. It is also possible to use predicted latents to reconstruct seen video clips at a low frame-per-second rate (see Figure S3), which we do not attempt in this study.

Table 1: **Within-subject metrics for all subjects and all latent types on the test set** Reported metrics are relative median rank ↓ (MR) of retrieval on a set of 500 samples, top-5 accuracy % ↑ (Acc) of retrieval on a set of 500 samples. These results were averaged across 50 retrieval sets, hence results are reported with a standard error of the mean (SEM) smaller than 0.01. The *Dummy* model systematically predicts the mean latent representation of the training set.

|  | CLIP 257 × 768 | | VD-VAE | | CLIP CLS | | AutoKL | |
|---|---|---|---|---|---|---|---|---|
|  | MR | Acc | MR | Acc | MR | Acc | MR | Acc |
| Dummy | 50.0 | 1.0 | 50.0 | 1.0 | 50.0 | 1.0 | 50.0 | 1.0 |
| S1 | 9.4 | 13.8 | 29.9 | 3.0 | 15.1 | 8.4 | 24.9 | 3.9 |
| S2 | 6.8 | 16.4 | 30.2 | 3.5 | 10.6 | 10.5 | 21.8 | 3.8 |
| S3 | 7.8 | 13.6 | 28.5 | 3.1 | 11.0 | 9.9 | 26.0 | 3.3 |

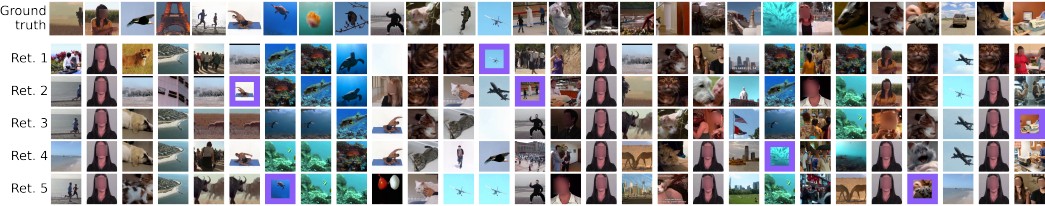

Figure 2: **Image retrievals using predicted latent representations of CLIP 257 × 768 latents** We use a model fitted on Subject 2 (S2) and predict the latent representation of unseen videos (test set). Ground truth images featured within the first 5 retrieved images are indicated with a bold purple border. In a given column, images which appear similar across rows are actually different frames of the same video clip. Images featuring human faces were blurred.

### 3.2 OUT-OF-SUBJECT DECODING AND MULTI-SUBJECT TRAINING

As illustrated in Figure 3, models trained on one subject do not generalise well to other subjects. However, we demonstrate that functional alignment can successfully be used as a transfer learning strategy to generalise a pre-trained model to left-out subjects. In particular, we show that left-out

subjects need not have the same amount of available data than training subjects to benefit from their model: with just 30 minutes of data, left-out subjects can reach performance which would have needed roughly 100 minutes of data in a within-subject setting. Besides, compared to the out-of-subject baseline, we obtain 25 to 75 percents improvement in relative median rank across latent types. Note that, in this study, we chose the best performing subject (S2) as the reference subject.

Finally, we show that a single model trained on all functionally aligned subjects can reach slightly better results than models trained on all un-aligned subjects. In every subject, this multi-subject aligned model performs comparably to their associated within-subject model. Supplementary Figures S5 and S6 show that these results hold for all types for latents.

Other interesting setups are reported in Figures S8, S9, S10, S11. In particular, they show that a multi-subject aligned model (e.g. trained on S1 and S2) has better performance on aligned left-out subjects (e.g. S3) than a single-subject model (e.g. trained on S2 only).

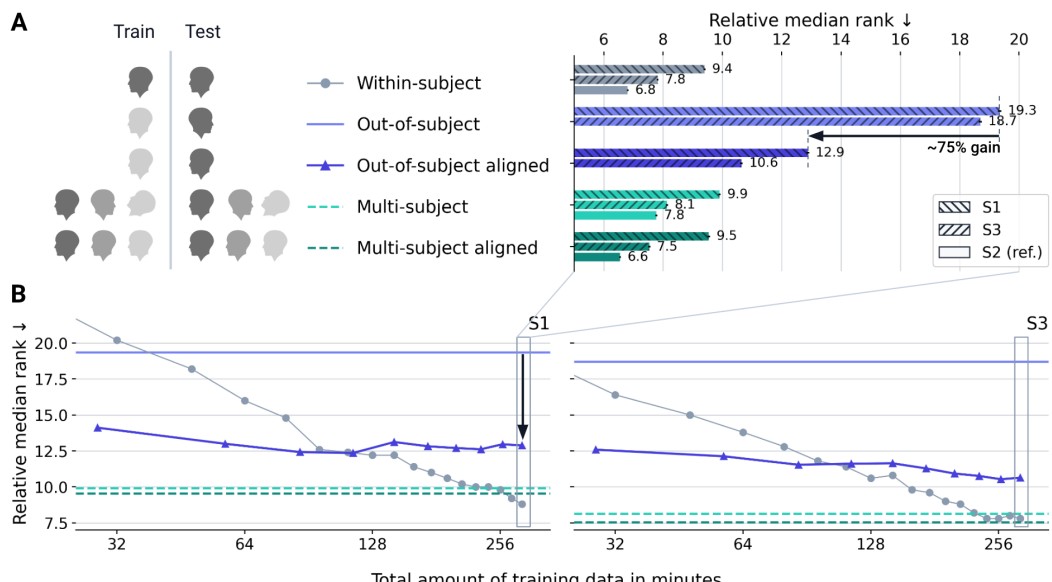

Figure 3: **Effects of functional alignment on multi-subject and out-of-subject setups**
We report relative median rank ↓ in all setups described in section 2.3 for CLIP $257 \times 768$. In all *aligned* cases, S1 and S3 were aligned onto S2. In all *out-of-subject* cases, we test S1 and S3 onto a decoder trained on S2. In all *multi-subject* cases, the decoder was trained on all data from all 3 subjects. **A.** In this panel, all models (alignment and decoding) were trained on all available training data. Results for other latent types are available in Figure S5. **B.** In left-out S1 and S3, decoding performance is much better when using functional alignment to S2 (solid dark purple) than when using anatomical alignment only (solid pale purple). Performance increases slightly as the amount of data used to align subjects grows, but does not always reach levels which can be achieved with a single-subject model fitted in left-out subjects (solid pale gray dots) when a lot of training data is available. Training a model on multiple subjects yields good performance in all 3 subjects (dashed pale teal) which can be further improved by using functional alignment (dashed dark teal). Results for other latent types are available in Figure S6.

To better understand how brain features are transformed by functional alignment, we show in Figure 4 how vertices from S1 are permuted to fit those of S2. Note that both subjects' data lie on fsaverage5. To this end, we colorize vertices in S1 using the MMP 1.0 atlas (Glasser et al., 2016) and use $\phi_{S1 \to S2}$ to transport each of the three RGB channels of this colorization. We see that, even in low data regimes, FUGW scrambles most of the brain but can leverage signal to recover the cortical organization of the occipital lobe. Higher regimes yield anatomically-consistent matches in a much higher number of cortical areas such as the temporal and parietal lobes, and more surprisingly in the primary motor cortex as well, while the prefrontal cortex and temporo-parietal junction (TPJ) still seem challenging to map.

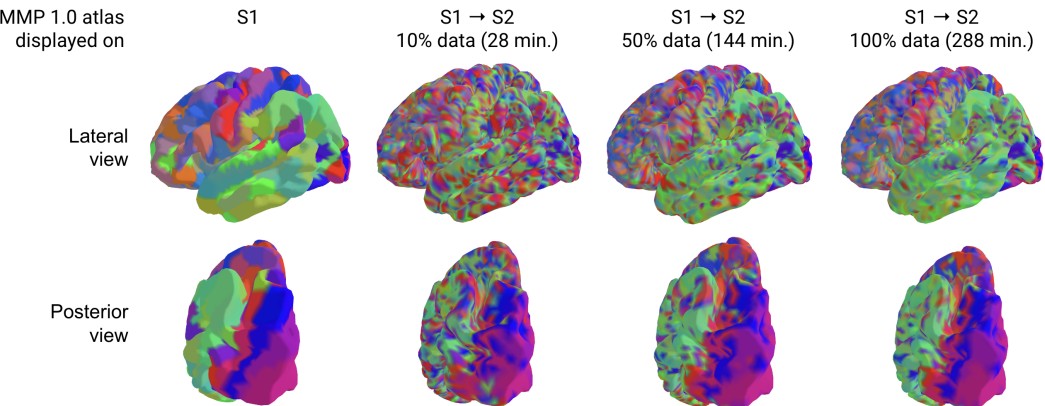

Figure 4: **Visualizing functional alignments in the left hemisphere** Vertices of the source subject (left) are permuted by FUGW. The result of this permutation is visualized on the target subject (columns 2, 3, and 4). Fitting FUGW with different increasing amounts of data gradually unfolds the cortical organisation of multiple areas, even non-visual ones. Note that all 3 models have been fitted using the same number of iterations.

### 3.3 INFLUENCE OF TRAINING SET SIZE AND TEST SET REPETITIONS

Recent publications in brain-decoding using non-invasive brain imagery show impressive results. However, we stress that these results are obtained in setups which are very advantageous when it comes to both dataset size and signal-to-noise ratio. To better assess the importance of these two factors, we report in Figure 5 performance metrics for subject models trained and tested with various amounts of data and various amounts of noise.

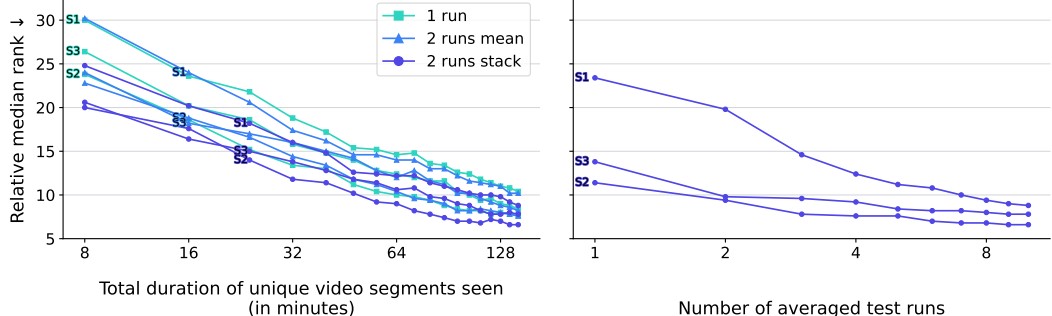

Figure 5: **Effect of training set size and test set noise on retrieval metrics** Relative median rank ↓ on a fixed test set gets better as more training data is used to fit the model (left). Interestingly, averaging brain volumes of 2 similar runs does not bring improvements compared to using just 1 run. Instead, stacking runs does yield significant improvements. Note that training sets using 2 runs have twice as much data as those using 1 run. Finally, these metrics are highly affected by the noise level of the test set (right): averaging more runs in the test set yields better metrics despite using the same decoder.

Firstly, using a fixed test set in which brain features were averaged across runs, we find that exponentially more training data is required per subject to achieve better performance. This finding is similar to that of systematic scaling studies on similar topics (Tang et al., 2023). More interestingly, in this given signal-to-noise ratio setup, it seems that more diverse training data should bring comparable or better performance than repeating already seen content, while potentially covering more semantic domains.

Secondly, reported performance metrics only hold in favorable signal-to-noise setups. Indeed, the test set associated with the Wen 2017 dataset comes with 10 runs for each video segment, which, when averaged together, greatly reduce the noise level. However, as reported here, when tested in real-life signal-to-noise conditions (i.e. only one run per video clip), our models' performance degrades: it is approximately twice as bad for each subject when using CLIP latents.

## 4 DISCUSSION

**Impact** The present work confirms the feasibility of using BOLD fMRI signal acquired in a naturalistic setup to decode high level visual features (Nishimoto et al., 2011). It further demonstrates that it is possible to leverage fMRI signal from naturalistic movie watching to derive meaningful functional alignments between subjects, which in turn can be used to transfer decoding models to novel subjects.

In particular, our study shows that decoding brain data from a left-out subject can be substantially improved by aligning this left-out subject to a large reference dataset on which a decoder was trained. Out method thus paves the way towards using models used on large amounts of individual data to decode signal acquired in smaller neuro-imaging studies, which typically record one hour of fMRI for each subject (Madan, 2022).

Besides, this study reports decoding accuracy in setups where subjects are showed test stimuli for the first time only, hence yielding insights on how these models would perform in real-time decoding. While performance improves with the number of repetitions at test time, reasonable decoding performance of semantics can be achieved in two out of three subjects with just one repetition.

Lastly, by systematically quantifying decoding accuracy as a function of the amount of training data, the present work brings insightful recommendations as to what stimuli should be played in future fMRI datasets collecting large amounts of data in a limited number of subjects. In the current setup (naturalistic movie watching at 3T), more diverse semantic content is more valuable than repeated content for fitting decoding models.

**Limitations** This work is a first step towards training accurate semantic decoders which generalize across individuals, but subsequent work remains necessary to ensure the generality of our findings.

Firstly, although reported gains in out-of-subject setups are significant, the small number of participants present in the dataset under study calls for replications on other – and potentially larger – cohorts. However, to our knowledge, no other dataset presented similar features to that of Wen et al. (2017) – i.e. high quantity of data per subject and large variety of video stimuli. The recent Courtois Neuromod dataset [5] might be useful in this regard.

Secondly, our approach currently requires left-out subjects to watch the same videos as reference subjects. It is yet unclear whether functional alignment could bring improvements without this constraint. However, multi-subject decoding can probably help partially address this issue: since it is possible to train a decoder on multiple subjects and because not all of them have to watch the same movies, it is possible that a lot of different movies could be used as "anchor" for left-out individuals.

Thirdly, unlike other approaches (Défossez et al., 2022), our approach relies on pre-trained encoders, and cannot align all subjects at once. Consequently, overall performance highly depends on the quality of other models and of data acquired in reference individuals.

Finally, while restricting this study to linear models makes sense to establish baselines and ensure reproducibility, non-linear models have proved to be very efficient. A natural improvement on this work could include these architectures.

**Ethical implications** Out-of-subject generalization is an important test for decoding models, but it raises legitimate concerns. In this regard, this study highlights that signal-to-noise ratio still currently makes it challenging to very accurately decode semantics in a real-time setup, and that a non-trivial amount of data is needed per individual for these models to work. Moreover, we stress that, while decoding perceived stimuli is making great progress, imagined stimuli are still very challenging (Horikawa & Kamitani, 2017). Nonetheless, it is important for advances in this domain to be publicly documented. We thus advocate that open and peer-reviewed research is the best way forward to safely explore the implications of inter-subject modeling, and more generally brain decoding.

**Conclusion** Overall, these results provide a significant step towards real-time, subject-agnostic visual decoding of semantics using fMRI.

---

[5] https://www.cneuromod.ca

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
