## A APPENDIX

### A.1 DATA PRE-PROCESSING

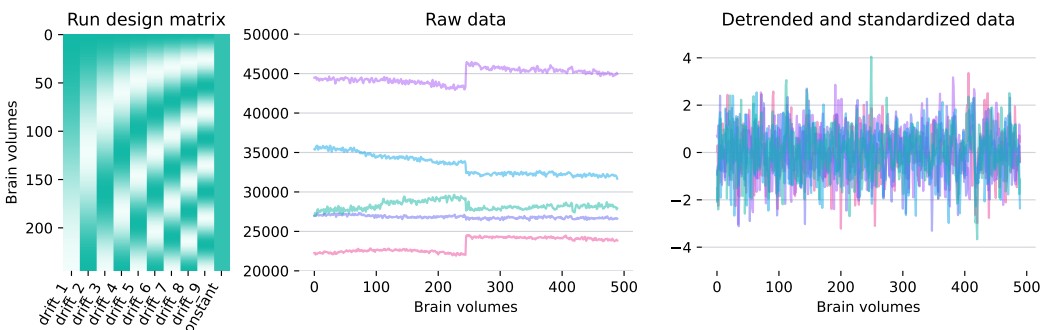

Figure S1: **Pre-processing of the Wen 2017 dataset** For each subject and each run, in each vertex, we regress out parts of the signal which can be linearly explained by the design matrix represented on the left, which models cosine drifts of the BOLD signal. The two graphs to the right show time-courses in 5 vertices across 2 different runs before (left) and after (right) they have been pre-processed.

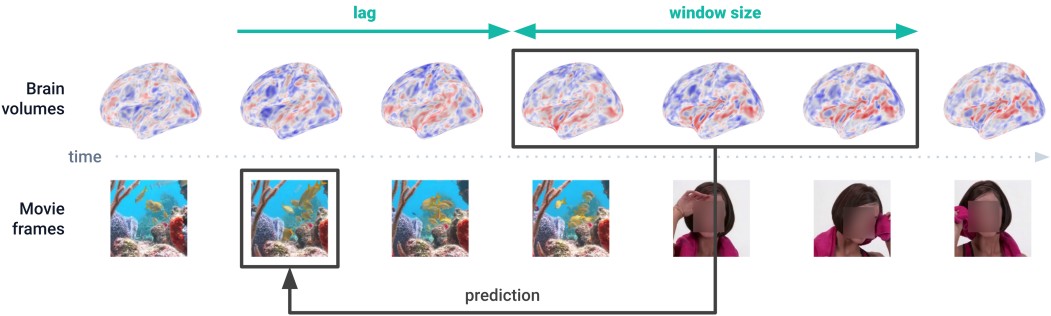

Figure S2: **Lag and window size** In order to decode a movie frame which was seen at time $t$, one can use brain volumes which were acquired further in time. This delay is referred to as the *lag*. Moreover, one can use several brain volumes to decode a given movie frame. The number of brain volumes used is called the *window size*. Images featuring human faces were blurred.

### A.2 RESULTS FOR EVERY TYPE OF LATENT REPRESENTATION

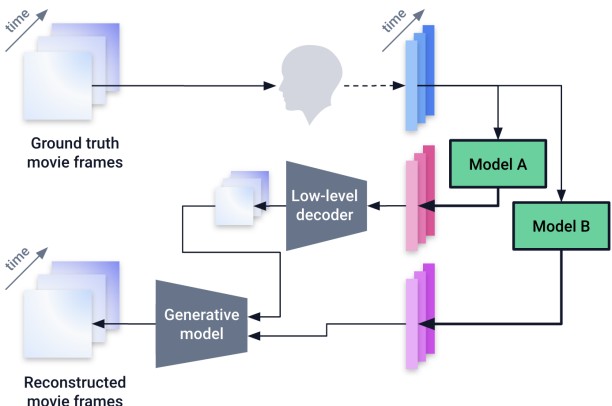

Figure S3: **Using predicted latents as inputs of generative models** Although out-of-scope for this study, it is possible to use predicted latents as inputs to one or multiple generative models to reconstruct visual stimuli seen by individuals.

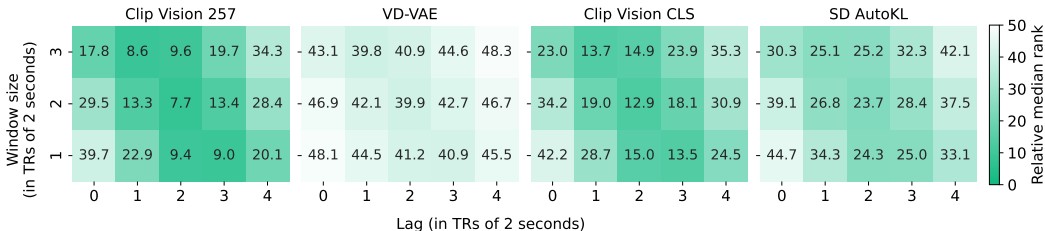

Figure S4: **Relative median rank ↓ of predicted latents averaged across subjects for various time lags and window sizes**

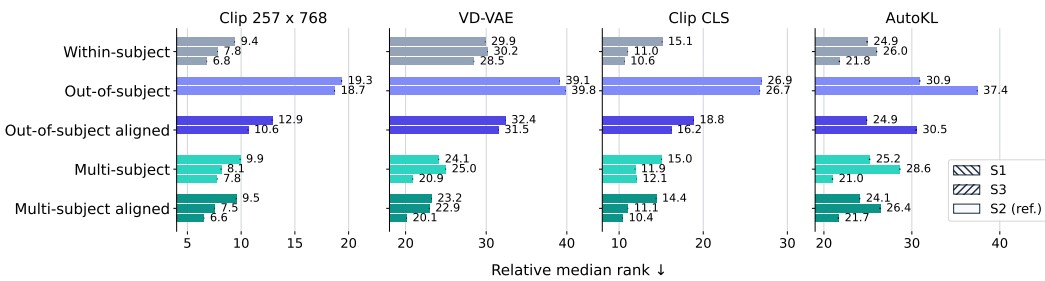

Figure S5: **Effects of alignment** For any type of latent representation, out-of-subject decoding performance, measured through relative median rank ↓, greatly improves when subjects are functionally aligned. Training decoders on multiple subjects also works better when subjects are aligned. These results were averaged across 50 retrieval sets ; all these metrics are reported with a standard error of the mean (SEM) smaller than 0.01.

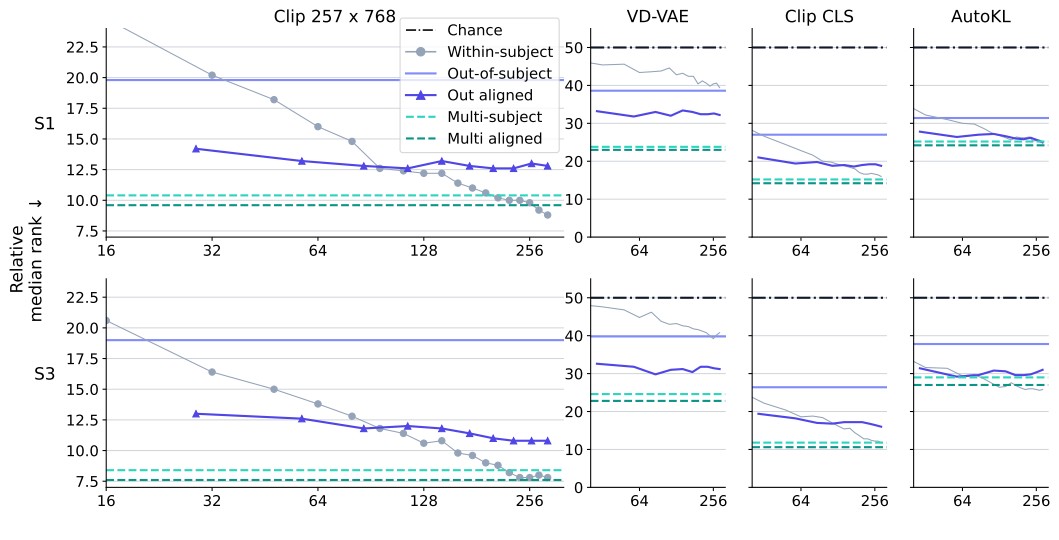

Figure S6: **Performance increases slightly with more alignment data** For any type of latent representation, out-of-subject decoding performance greatly increases with functional alignment even in low data regimes. In high data regimes, out-of-subject decoding does not work as well as fitting single-subject or multi-subject models.

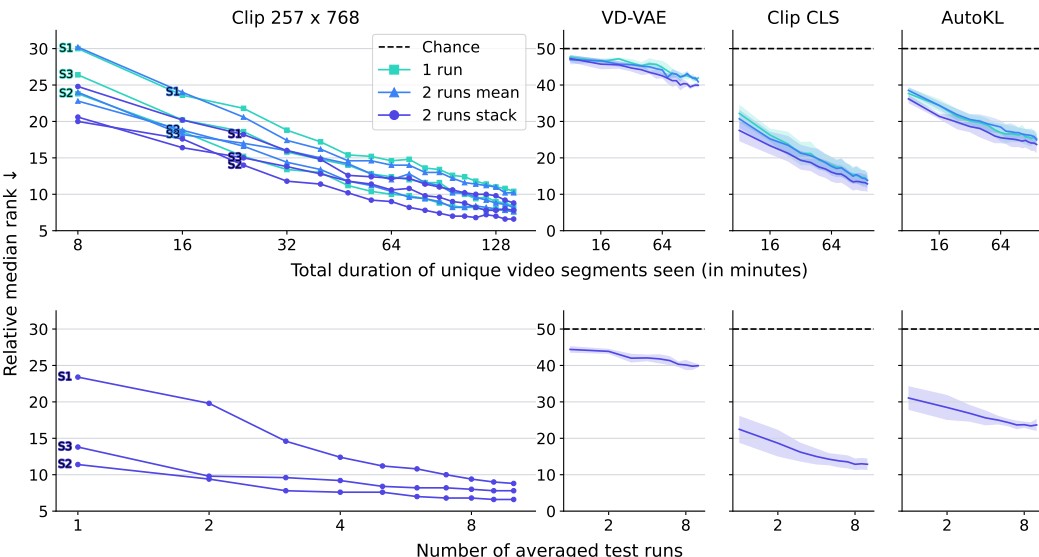

Figure S7: **Scaling studies for all latents** For any type of latent representation, decoding performance increases linearly with exponentially more data. It also seems that, when acquiring data at 3T or more, not repeating stimuli yields the best results. At test time, although repeating stimuli allows to get better metrics, retrieval performance with only one repetition is already reasonable in 2 out of 3 subjects of the Wen 2017 dataset.

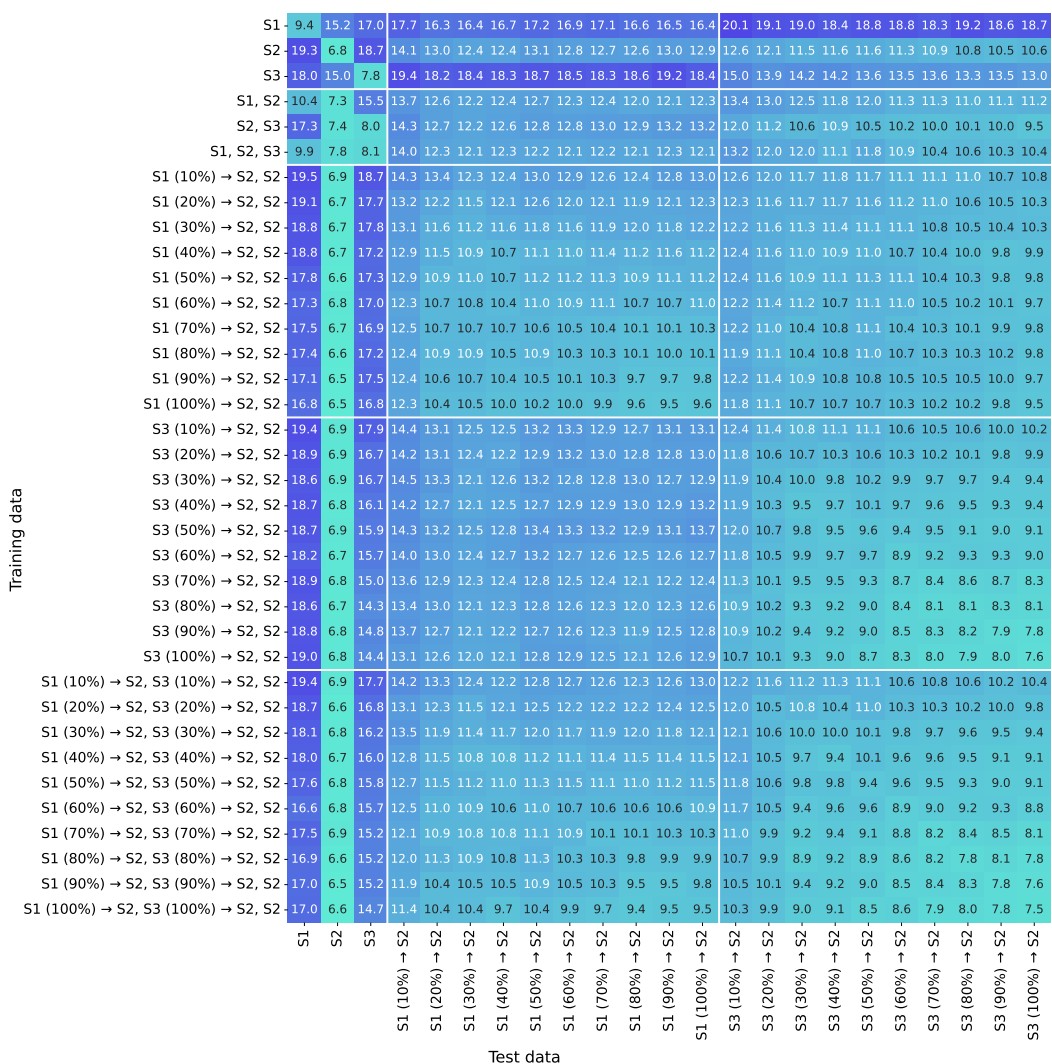

Figure S8: Relative median rank ↓ for **CLIP 257 × 768** latents in single- and multi-subject training sets, with and without alignment, tested on within- and across-subjects setups with and without alignment. These results were averaged across 50 retrieval sets ; all these metrics are reported with a standard error of the mean (SEM) smaller than 0.01.

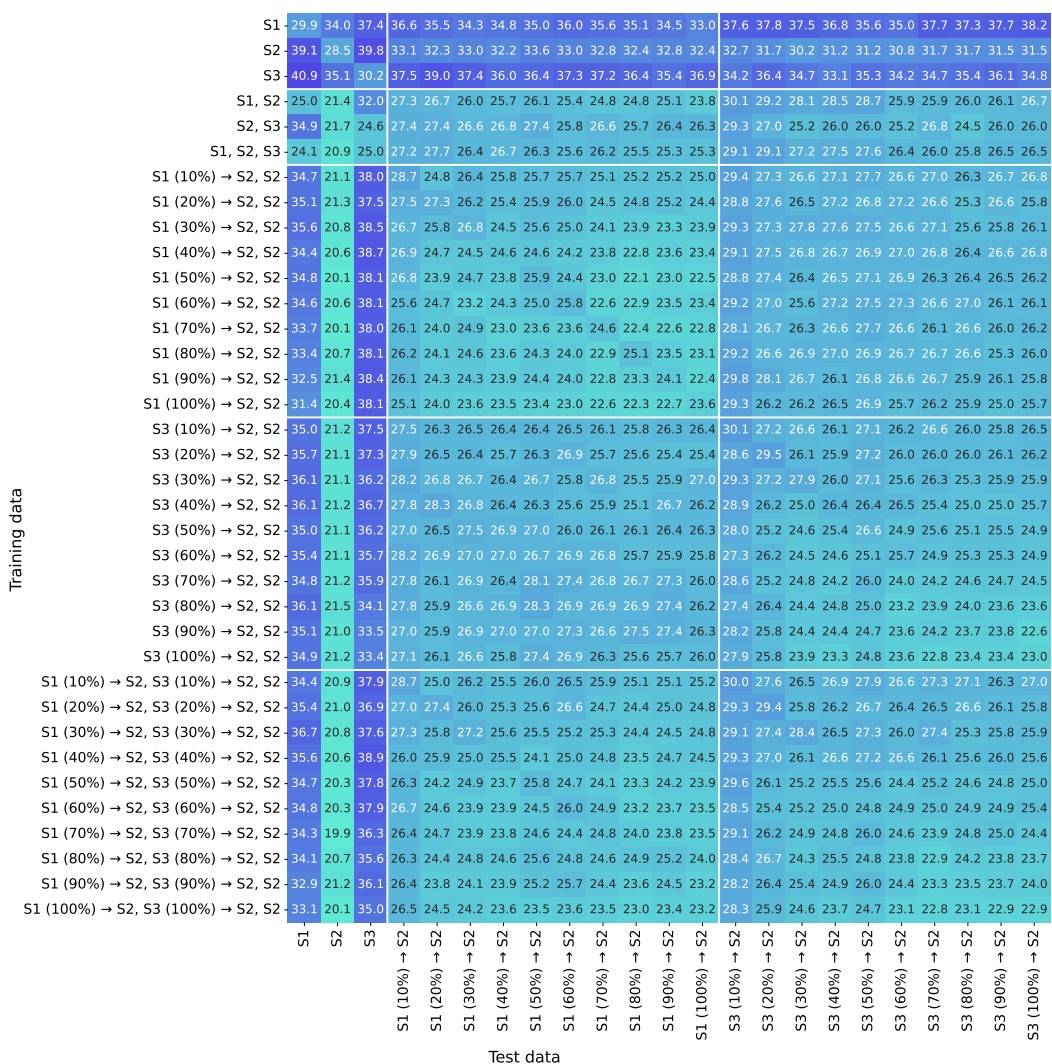

Figure S9: Relative median rank ↓ for **VD-VAE** latents in single- and multi-subject training sets, with and without alignment, tested on within- and across-subjects setups with and without alignment. These results were averaged across 50 retrieval sets ; all these metrics are reported with a standard error of the mean (SEM) smaller than 0.01.

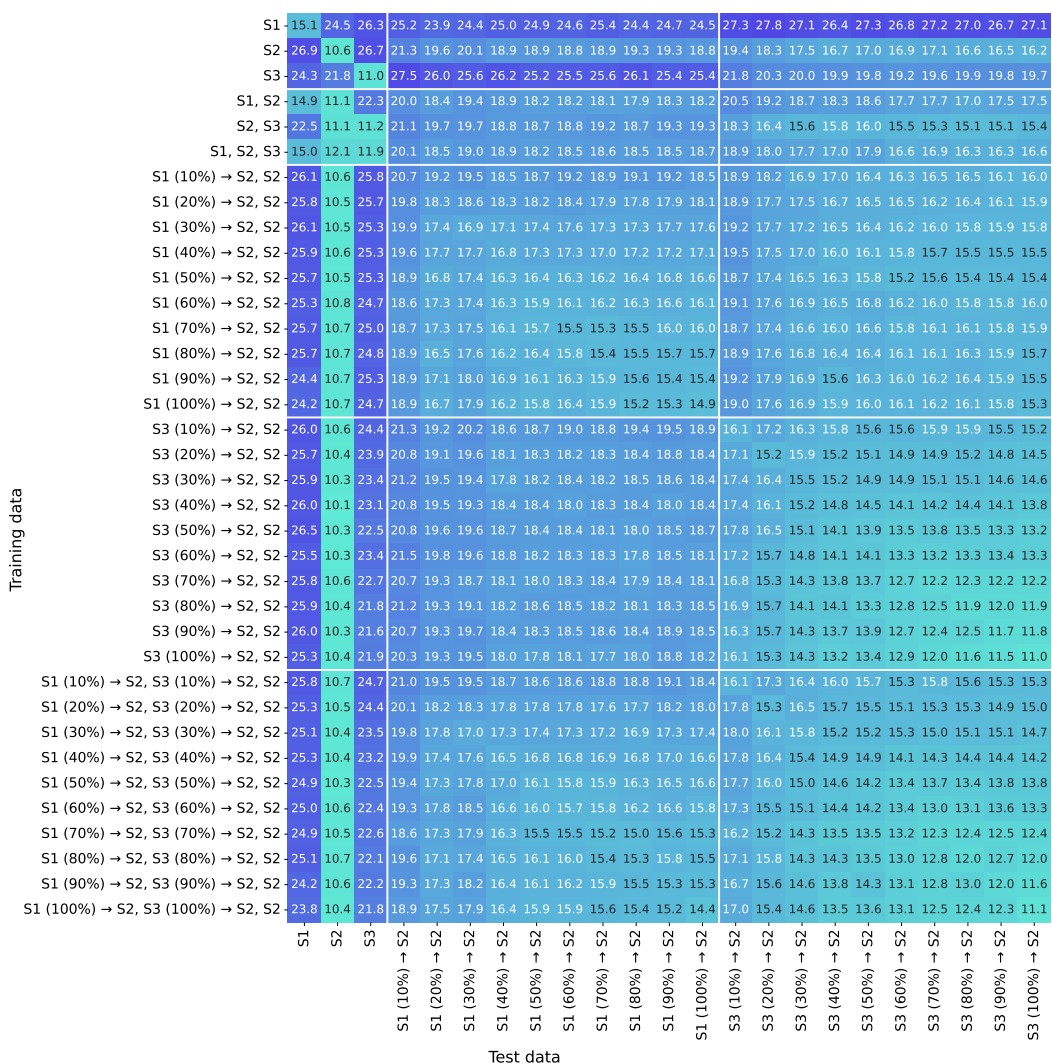

Figure S10: Relative median rank ↓ for **CLIP CLS** latents in single- and multi-subject training sets, with and without alignment, tested on within- and across-subjects setups with and without alignment. These results were averaged across 50 retrieval sets ; all these metrics are reported with a standard error of the mean (SEM) smaller than 0.01.

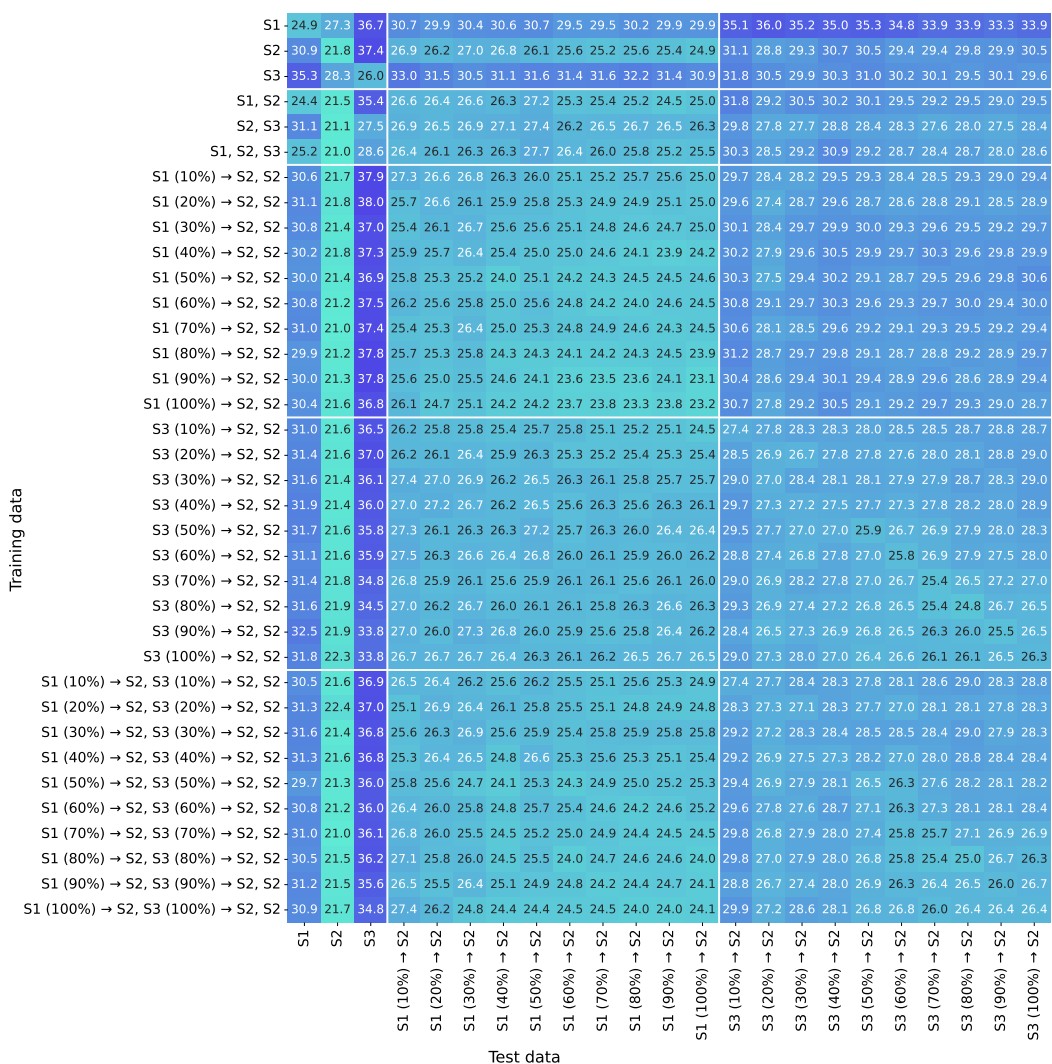

Figure S11: Relative median rank ↓ for **AutoKL** latents in single- and multi-subject training sets, with and without alignment, tested on within- and across-subjects setups with and without alignment. These results were averaged across 50 retrieval sets ; all these metrics are reported with a standard error of the mean (SEM) smaller than 0.01.