# OpenReview forum: "Aligning brain functions boosts the decoding of videos in novel subjects"
_ICLR.cc/2024/Conference — Submitted to ICLR 2024_

### Official Review · Reviewer_jWwB · 2023-10-30

**Soundness:** 1 poor
**Presentation:** 1 poor
**Contribution:** 2 fair
**Rating:** 3
**Confidence:** 4

**Summary:**

This paper proposed a aligment method to boost the performance of brain decoding.

**Strengths:**

This paper proposed a simple method for brain decoding.

**Weaknesses:**

1.The loss function has three parameters. How to choose the parmeters is difficult.
2.The proposed method should be compared with the state-of-the-art methods.

**Questions:**

1. The dataset is very small. Hence, How is the generalization of the model？
2. The proposed method don't compared with the state-of-the-art methods.

---

> ### Author Response · Authors · 2023-11-14
>
> We thank the reviewer for their time and feedback.
>
> We want to stress that we respectfully disagree with the reviewer’s summary of our work. Our manuscript does not propose an alignment method. It instead shows that at least one functional alignment method (introduced in a different paper) is suited to transfer semantic decoders trained in some individuals to other, left-out individuals. It is possible that other functional alignment methods could achieve better performance in this regard, although we believe it is out of the scope of this study to compare them. Also, we stress that anatomical alignment (i.e. projecting all data to a common anatomical template like MNI or fsaverage) is the go-to strategy employed in a vast majority of studies. Therefore, since this anatomical alignment is our baseline, our work already includes an insightful comparison to the most commonly used alignment method in the literature. Finally, as stated in Bazeille et al. 2021, it is not clear that any existing functional alignment method outperforms others in the general case (in particular, performance highly depends on the downstream task and input data).
>
> **Weaknesses**
>
> 1. Reviewer jWwB is right that the chosen alignment method (FUGW) has 3 hyper-parameters, namely alpha, rho and epsilon. We use FUGW because, unlike alternatives, it comes with a simple python API comparable to that of sklearn and thus facilitates reproducibility. Previous work has already explored the impact of these three hyper-parameters, and showed that alignment is robust to changing values for alpha and rho, although epsilon needed to be carefully chosen. Our goal is a proof-of-existence, i.e. that it is possible to yield significant decoding gains with a standard functional alignment preprocessing.
> 2. Since we did not introduce this alignment method, we did not feel compelled to compare it with a lot of other methods. However, note that our baseline (i.e. fsaverage) is the most commonly used alignment method. Moreover, we agree with you that our work could constitute an interesting downstream task, which a future benchmark could use to compare between functional alignment methods.
>
> **Questions**
>
> 1. Indeed, fMRI datasets featuring a lot of data per participant are still very rare. To our knowledge, the Wen 2017 featured in our study is among the most extensive datasets in this regard. In order to strengthen our work, we have decided to implement two other experiments, which reproduce our analysis with each possible reference subject. We kindly invite you to read their description in the response we make to reviewers uqYK and HiZ5.
> 2. We hope our second point of the previous section addresses your concern.
>
> We thank jWwB for their helpful remarks, and hope these comments help alleviate their concerns. We would be happy to provide additional information and experiments which could help strengthen our conclusions.

---

> > ### Author Response · Authors · 2023-11-18
> >
> > Let us further expand on our previous answer by providing results to the two additional experiments described in our previous message, which we think bring insight on the robustness of our finding. We kindly invite you to read them in the second comment we have addressed to reviewer uqYK.
> >
> > These two experiments bring new information regarding your first question. In short, the first experiment shows that our results still hold regardless of which individual is used as the reference subject. Moreover, the second experiment shows that training on multiple aligned subjects can help boost results of our out-of-subject setup even further.
> >
> > We hope that these new elements will strengthen our results and would happily provide additional information.

---

> > > ### Author Response · Authors · 2023-11-22
> > >
> > > In order to further address your first question, we replicated our main experiment on the Natural Scenes Dataset. We find results similar to that exposed in our paper, i.e. that functional alignment dramatically boosts the performance of semantic decoders when they are used on left-out subjects. We kindly invite you to read them in the second comment we have addressed to reviewer HiZ5.
> > >
> > > We hope that these new elements strengthen our current results. We will integrate them in our manuscript.

---

### Official Review · Reviewer_tcU3 · 2023-10-31

**Soundness:** 3 good
**Presentation:** 3 good
**Contribution:** 2 fair
**Rating:** 3
**Confidence:** 3

**Summary:**

This paper proposes to boost the decoding of videos in a single left-out subject with an alignment model and a decoder. The alignment model includes anatomical alignment and functional alignment. Finally, the pre-trained image encoder is used to obtain the video output.

**Strengths:**

The proposed ideas are simple and intuitive. It's quite generic and can be applied to different models.

**Weaknesses:**

1. The novelty needs to be further elaborated.
1. The proposed method lacks a comparison with other models. As a result, the effectiveness of the method is not convincing and requires further validation.
2. The proposed method in this paper employs one reference subject. How to deal with multiple reference subjects in practice?

**Questions:**

Please refer to the weakness.

---

> ### Author Response · Authors · 2023-11-14
>
> We thank the reviewer for their time and feedback.
>
> **Weaknesses**
>
> 1. We agree with the reviewer that we should clarify the novel contributions of our work. We propose to amend the introduction  with the following paragraph:
> “Overall, decoding and functional alignment studies have focused on largely disjoint objectives: decoding semantics is typically done at the individual level (e.g. Tang et al. 2023, Scotti et al. 2023, Ozcelik et al. 2023), whereas functional alignment across subjects is evaluated on simple features, such as retinotopy (Feilong et al. 2022), object categorization (Bazeille et al. 2021), and binary contrasts (Thual et al. 2022). With the rise of naturalistic datasets (NSD, Wen2017, Lebel2023, Things), it is pressing to know the conditions necessary for future studies to directly benefit from the high-level representations identified in a small set of high-quality recordings. Here we systematically investigate the data regime in which a set of standard semantic decoders would benefit from functional alignment.”
> 2. Reviewer tcU3 is right that we only benchmark the functional alignment method (FUGW) against the most commonly used anatomical alignment method, which is our baseline. FUGW was introduced and benchmarked in a different paper. The goal of the present study is to evaluate the condition in which functional alignment could improve standard  semantic decoders. Moreover, the baseline used in our paper is the alignment method most commonly used in neuroscience.
> 3. We agree with tcU3 that the present method is limited to a single reference subject. To mitigate this issue, we propose to generalise our result section, by reproducing the analysis with each possible reference subject. We kindly invite you to read the description of two additional experiments in the response we make to reviewers uqYK and HiZ5. Overall, however, building a multi-subject reference is still an open research question. Thual et al. 2022 introduces a method to build functional barycenters of individuals, which we did not try here because we feel that this way of computing reference subjects is fairly new and would require extensive testing of its own.
>
> We thank tcU3 for their helpful remarks, and hope these comments help alleviate their concerns. We would be happy to provide additional information and experiments which could help strengthen our conclusions.

---

> > ### Author Response · Authors · 2023-11-18
> >
> > Let us further expand on our previous answer by providing results to the two additional experiments described in our previous message, which we think bring insight on the robustness of our finding. We kindly invite you to read them in the second comment we have addressed to reviewer uqYK.
> >
> > These two experiments bring new information regarding your third question. In short, the first experiment shows that our results hold regardless of which individual is used as the reference subject. Moreover, the second experiment shows that training on multiple aligned subjects can help boost results of our out-of-subject setup even further. It also shows that any subject can be used as reference to train the decoder and alignment.
> >
> > We hope that these new elements will strengthen our results and would happily provide additional information.

---

> > > ### Author Response · Authors · 2023-11-22
> > >
> > > In order to further elaborate on the point raised by your third question, we replicated our main experiment on the Natural Scenes Dataset. This replication shows results similar to that exposed in our paper, i.e. that functional alignment dramatically boosts the performance of semantic decoders when they are used on left-out subjects. They also show that more data will be needed to figure out a principled way to choose or build the reference subject. We kindly invite you to read them in the second comment we have addressed to reviewer HiZ5.
> > >
> > > We hope that these new elements strengthen our current results. We will integrate them in our manuscript.

---

### Official Review · Reviewer_HiZ5 · 2023-10-31

**Soundness:** 3 good
**Presentation:** 3 good
**Contribution:** 3 good
**Rating:** 6
**Confidence:** 3

**Summary:**

The work deals with the decoding of high-level visual features from fMRI recordings. The authors use functional alignment to align fMRI data across subjects. The work claims that using functional alignment instead of standard structural methods boosts the decoding performance when there is limited data available for the subject in case. Further, training a model with aligned subjects can be used, whereas previous models could only decode responses specific to a subject.

**Strengths:**

Using functional alignment to improve decoding of visual representations across subjects is novel and noteworthy. The quantification of decoder performance with respect to data size is noteworthy.

**Weaknesses:**

The approach of showing the left-out subject the same video as the reference subject is a substantial weakness. This coupled with the functional alignment could theoretically act as a “leakage” mechanism for the data

**Questions:**

Does one repetition in line 268 mean the first trail or the second?

Line 281 says the left-out subjects have to watch the same videos, while, line 265 says the subjects are shown the test stimuli for the first time. Does this mean the videos shown themselves are new to the subjects?

Where is it shown that the approach aligns brain responses in accordance with brain anatomy? ( LIne 12)

Perhaps, adding at least a single subject comparison where the subject is shown different stimuli may help have more robust results

---

> ### Author Response · Authors · 2023-11-14
>
> We thank the reviewer for their time and feedback.
>
> **Weaknesses**
>
> In our setups, the test set is completely separated from the training sets of both the decoder and alignment: the training and test sets contain different movies acquired during different MRI sessions.
>
> However, reviewer HiZ5 is right that our alignment (although not the decoder) requires the participants to watch the same stimuli. This limitation applies to the vast majority of alignment methods (Bazeille et al. 2021).
>
> To clarify this issue we propose to indicate the following in the method section:
> “For example, in the out-of-subject setup, a decoder is trained on movies 1, 2, 3 seen by subject A. Subjects A and B have both seen movies 4 and 5, which are used to compute an alignment. We refer to these movies as “anchors”, because they allow left-out subjects to “anchor” themselves onto a decoder that has been trained on another individual. Finally, the decoder is tested on data from subject B, acquired when they were watching movie 6. All aforementioned movies were acquired in different MRI sessions.”
>
> In our paper, movies used to train the decoder and alignment are the same, which does not need to be true in the general case. However, there is no reason not to include “anchor” movies in the training dataset of the decoder, as it makes for more accurate decoders and more powerful “anchors” to the decoder.
>
> **Questions**
>
> 1. In this case, we used only the first trial (i.e. run). We could try to use only the second run, but we think it is more crucial to measure results based on data which represents the neural response of individuals when they are first shown the stimuli. Indeed, stimuli repetition can lead to unexpected effects (drop of attention, less controlled experiment in general).
> 2. We hope that our response to the main weakness answers your question. Please let us know if further clarification would be helpful.
> 3. We further expand on our claim that alignments map vertices in accordance with brain anatomy: in Figure 4, the first column from the left shows an atlas displayed on subject 1, without functional alignment (in this case, using only anatomical alignment). Columns 2, 3, and 4 show how vertices of subject 1 are reorganised on the cortical sheet to functionally match that of subject 2, using an increasing amount of movie-watching data to fit the functional alignment. What we observe is that, with more data, computed reorganisations look more anatomically coherent. For instance, the parietal and temporal lobe look very scrambled in column 2 (i.e. some of their vertices get matched with vertices coming from very different cortical areas), but much more consistent in column 4 (based on colour, matched vertices are much more coherent). One can observe a similar effect even in premotor areas, and the occipital lobe.
> We are working on a refined version of this Figure to make these points clearer.
> 4. We hope that our response to the main weakness answers your question. Please let us know if further clarification would be helpful.
>
> Again, we are thankful to HiZ5 for these valuable comments.

---

> > ### Author Response · Authors · 2023-11-22
> >
> > In order to further address your point regarding data leakage, we replicated our experiment on the Natural Scenes Dataset, while completely separating the dataset used to train the decoders from the dataset used to train the alignments. We find similar results to those that are exposed in our paper.
> >
> > Here is how we built our dataset:
> > - We only worked with subjects who had completed all trials (i.e. subjects 1, 2, 5 and 7)
> > - Each subject is shown 10 000 images 3 times ; among these, 1 000 images are shown to all participants
> > - We use all trials for these 1 000 images to compute functional alignments between each pair of individuals
> > - We use all trials for 8 000 / 9 000 of individual images to train a decoder in each participant, and the all trials for the remaining 1 000 / 9 000 images to test these decoders
> >
> > We report the Relative Median Rank (lower is better, chance level is at 50, and 0 means perfect accuracy) for decoders trained in 1 individual (the reference subject) and tested on another unaligned / aligned individual (the left-out subject). We see that, in all possible combinations, functional alignment yields significant decoding gains compared to anatomical alignment. In this setup, stimuli used to (1) train the decoder, (2) compute the alignment and finally (3) test the decoder are all disjoint sets.
> >
> > | Reference subject | Left-out subject | Not aligned | Aligned  |
> > | ----------------- | ---------------- | ----------- | -------- |
> > | S1            	| S2           	| 28.3    	| **15.1** |
> > | S1            	| S5           	| 33.4    	| **12.4** |
> > | S1            	| S7           	| 36.1    	| **18.1** |
> > | S2            	| S1           	| 24.4    	| **14.5** |
> > | S2            	| S5           	| 30.8    	| **12.7** |
> > | S2            	| S7           	| 32.1    	| **20.0** |
> > | S5            	| S1           	| 33.8    	| **14.7** |
> > | S5            	| S2           	| 31.4    	| **12.1** |
> > | S5            	| S7           	| 34.4    	| **16.5** |
> > | S7            	| S1           	| 32.3    	| **11.6** |
> > | S7            	| S2           	| 28.6    	| **13.9** |
> > | S7            	| S5           	| 30.4    	| **10.2** |
> >
> > For clarity, we also report the RMR when tested on data from the same individual it’s been trained on:
> >
> > | Subject | RMR |
> > | ------- | --- |
> > | S1  	| 4.6 |
> > | S2  	| 6.9 |
> > | S5  	| 4.3 |
> > | S7  	| 5.5 |
> >
> > We hope that these new elements strengthen our current results. We will integrate them in our manuscript.

---

### Official Review · Reviewer_uqYK · 2023-11-09

**Soundness:** 2 fair
**Presentation:** 3 good
**Contribution:** 2 fair
**Rating:** 5
**Confidence:** 3

**Summary:**

This paper is aimed at improving subject video decoding from BOLD functional responses obtained during a movie watching paradigm. To this end, the authors perform brain response alignment via an optimal transport methodology, followed by a linear regression to predict latent representations of video frames (obtained by standard encoder models such as CLIP or VD-VAE).  They find that this method improves out of subject video decoding performance when compared to a purely anatomical alignment approach. They also examine multi-subject alignment in comparison to single subject approaches when limited number of paired recordings are available

**Strengths:**

The paper provides some interesting insights into the use of functional alignment models for studying BOLD responses in the context of naturalistic stimuli. Although the alignment methodology is not entirely a novel contribution in and of itself, the experimental setup is well deigned to examine the hypotheses being tested. The findings are clearly presented and motivated

**Weaknesses:**

The major weakness of this paper is the limited number of subjects that are available for testing. Given that only three different subjects are used, it is unclear whether the findings of the paper and insights will generalize for a larger population.

**Questions:**

1. It is not clear how sensitive the method to the choice of image latent representation/granuarity of features? How was the choice of representational models (such as CLIP or VD-VAE etc) made, is one or the other more suitable for this evaluation setup? Additionally, is there a reason the video frame decoding is restricted to a linear regression parameterization

2. How was the choice of retrieval metric made? Is there a reason standard metrics such as mean average precision, or NDCG are not appropriate for evaluating this task?

3. It would be nice to provide more context to explaining the design and modeling strategy in Eq. (1). The way it is currently presented requires the reader to flip back and forth between this manuscript and Thual et al 2022 to understand the methodology properly.

4. It would be nice if a higher resolution image for Figure 2 could be made available to understand the false retrieval cases. Additionally, it would be nice to provide more description in the appendix to help the reader interpret the extended experimental observations.

---

> ### Author Response · Authors · 2023-11-14
>
> We thank the reviewer for their time and valuable feedback.
>
> **Weaknesses**
>
> We agree that it is unclear whether our multi-subject alignment will scale to larger numbers of individuals.
> Our decision to focus on a dataset with few individuals with a lot of natural stimuli stems from a simple question: under what data regime can functional alignment be beneficial to semantic decoding.
> In this regard, our contribution is a proof-of-concept: if decoders trained on a single participant can generalize to other unseen individuals for whom little data has been acquired, then it constitutes (1) a strong incentive to acquire very high amounts of data in a limited number of participants and (2) an additional legitimation of recent initiatives (e.g. Natural Scene Dataset, Kay, Naselaris et al. 2023; deep-fMRI-dataset, Lebel, Huth et al. 2023).
>
> To further address this issue, we propose to add two more experiments:
> - Experiment 1: evaluate decoding gains on each possible reference subject of the Wen2017 dataset. This will allow us to triple the number of experiments, hopefully strengthening our original conclusions.
> - Experiment 2: evaluate decoding gains for out-of-subject setups in which the decoder has been trained on multiple aligned subjects.
>
> We will test all possible combinations to measure the effect of multi-subject training on this cohort.
> We will post results for both these experiments as soon as we get them.
>
> **Questions**
>
> 1. We chose to decode CLIP Vision 257 x 768 (high-level, semantic features) and VD-VAE (low-level features) latents to compare our results to that of Ozcelik et al. 2023 (in which the authors trained single-subject linear decoders on fMRI data from the Natural Scenes Dataset with much higher signal-to-noise ratio). We also included CLIP Vision CLS and AutoKL latents because they are comparable to CLIP Vision 257 x 768 and VD-VAE latents respectively, while being much smaller. We propose to clarify these choices in the “Video output” section.
> Supplementary figures S5-S7 illustrate the differences in decoding performance between these latents. In particular, we see that CLIP-based features (i.e. high level) are much more easily decoded than other, low-level features.
> Do you wish for us to make these results more clearly accessible in the main paper?
> The decision  to use linear decoder follows a classic approach in fMRI (e.g. Naseralis et al 2011, Huth et al Nature 2016, Ozcelik et al. 2023) which present multiple advantages: (1) simplicity, (2) interpretability and (3) robustness. Furthermore, our internal testing did not show significant improvement of deep-learning pipelines over linear approaches. We propose to clarify this motivation in the method section.
> 2. We chose to report retrieval metrics to be comparable with other recent papers (e.g. Scotti et al. 2023) and because image reconstruction metrics (like used in Ozcelik et al. 2023) depend on the generative models used, but are also difficult to assess. Also, top-k accuracy is used in similar work (Scotti et al. 2023; Defossez et al. 2023; Chen et al 2022). Finally, we think that relative median ranks are easier to understand intuitively and to compare between settings.
> 3. We agree that this part of the paper is confusing. We propose to rearrange the method sections to avoid this back-and-forth. We agree and will make a higher-resolution version of Figure 2 in the appendix and add more comprehensive captions and paragraphs to introduce supplementary results.
>
> Again, we are thankful to uqYK for these valuable comments.

---

> > ### Author Response · Authors · 2023-11-18
> >
> > Let us further expand on our previous answer by providing results to the two additional experiments described in our previous message, which we think bring insight on the robustness of our finding.
> >
> > **Experiment 1:** in this experiment, we seek to show that functional alignment allows to boost the performance of semantic decoders in left-out individuals, regardless of what individual is used as the reference individual. We used all available training data to train the decoder and compute the alignment. Test data is independent (different videos shown in different runs) from training data.
> >
> > We report the Relative Median Rank (lower is better) for decoders trained in 1 individual (the reference subject) and tested on another unaligned / aligned individual (the left-out subject). We see that, in all possible combinations, functional alignment yields significant decoding gains compared to anatomical alignment.
> >
> > | Reference subject | Left-out subject | Not aligned | Aligned  |
> > | ----------------- | ---------------- | ----------- | -------- |
> > | S2            	| S1           	| 19.3    	| **12.5** |
> > | S2            	| S3           	| 18.7    	| **10.0** |
> > | S1            	| S2           	| 15.2    	| **9.9**  |
> > | S1            	| S3           	| 17.0    	| **10.3** |
> > | S3            	| S1           	| 18.0    	| **13.8** |
> > | S3            	| S2           	| 15.0    	| **10.6** |
> >
> > We see that this transfer strategy yields performance close to that of decoders trained directly on left-out subjects, which we recall here:
> >
> > | Subject | RMR |
> > | ------- | --- |
> > | S1  	| 9.4 |
> > | S2  	| 6.8 |
> > | S3  	| 7.8 |
> >
> > Moreover, although this would need more extensive testing, we speculate at this stage that subjects whose individual decoder yields the best performance will make the best reference subjects (note how S2 and S3 yield better performances than S1 when used as reference subjects).
> >
> > **Experiment 2:** in this experiment, we seek to show that training a decoder on multiple subjects brings additional gains compared to the experiment above. Similarly, we used all available training data to train the decoder and compute the alignment. Test data is independent (different videos shown in different runs) from training data.
> >
> > Here, we train a decoder on unaligned / aligned pairs of individuals, and test it on a left-out individual. We report the Relative Median Rank in the following three tables.
> >
> > Let us consider the first table. The first line of this table reports results for which a decoder has been trained on S1 and S2 without alignment, and tested on S3 without alignment. The second line reports results for which a decoder has been trained on S1 and S2 without alignment and tested on S3 ➔ S1 (i.e. S3 aligned onto S1). The fourth line reports results for which a decoder has been trained on S1 and S2 ➔ S1 and tested on S3 ➔ S1.
> >
> > We observe two effects: (1) functional alignment yields systematic gains compared to anatomical alignment in multi-subject left-out subject setups and (e.g. RMRs are better for lines 4 and 5 than for line 1) and (2) aligned multi-subject yields gains compared to single-subject (e.g. S3 ➔ S2 is better decoded by a model trained with two subjects - line 5 of the first table below - than by a model trained on S2 only - line 2 of experiment 1’s main table). In order to avoid back-and-forth reading, we report results from experiment 1 in the last column of the following tables.
> >
> > | Training data | Test data | RMR 	| RMR Exp 1 |
> > | ------------- | --------- | ------- | --------- |
> > | S1, S2    	| S3    	| 15.5	| -     	|
> > | S1, S2    	| S3 ➔ S1   | 11.1	| -     	|
> > | S1, S2    	| S3 ➔ S2   | 10.7	| -     	|
> > | S1, S2 ➔ S1   | S3 ➔ S1   | 10.4	| **10.3**  |
> > | S1 ➔ S2, S2   | S3 ➔ S2   | **9.3** | 9.9   	|
> >
> > | Training data | Test data | RMR 	| RMR Exp 1 |
> > | ------------- | --------- | ------- | --------- |
> > | S1, S3    	| S2    	| 14.5	| -     	|
> > | S1, S3    	| S2 ➔ S1   | 10.4	| -     	|
> > | S1, S3    	| S2 ➔ S3   | 10.5	| -     	|
> > | S1, S3 ➔ S1   | S2 ➔ S1   | **9.8** | 9.9   	|
> > | S1 ➔ S3, S3   | S2 ➔ S3   | **9.6** | 10.6  	|
> >
> > | Training data | Test data | RMR  	| RMR Exp 1 |
> > | ------------- | --------- | -------- | --------- |
> > | S2, S3    	| S1    	| 17.3 	| -     	|
> > | S2, S3    	| S1 ➔ S2   | 13.0 	| -     	|
> > | S2, S3    	| S1 ➔ S3   | 13.2 	| -     	|
> > | S2, S3 ➔ S2   | S1 ➔ S2   | **12.0** | 12.5  	|
> > | S2 ➔ S3, S3   | S1 ➔ S3   | **12.9** | 13.8  	|
> >
> > We hope that these new elements will strengthen our results and would happily provide additional information.

---

> > > ### Author Response · Authors · 2023-11-22
> > >
> > > In order to further address the weakness you rightly pointed out, we replicated our main experiment on the Natural Scenes Dataset. We find results similar to that exposed in our paper, i.e. that functional alignment dramatically boosts the performance of semantic decoders when they are used on left-out subjects. We kindly invite you to read them in the second comment we have addressed to reviewer HiZ5.
> > >
> > > We hope that these new elements strengthen our current results. We will integrate them in our manuscript.

---

### Meta-Review · Area_Chair_ExXb · 2023-12-09

**Metareview:**

Reviewers wanted significant new experiments to explore how the results generalize to new datasets and to conditions that have more than 3 test subjects. Authors did provide such results in the responses, such as results on NSD, but these experiments cannot be understood through a brief description and results table in a short author response. Experiments, of the scale those provided in the original manuscript, require significant explanations and details in the form of an updated manuscript. Reviewers have no way of evaluating such results, even if the initial author responses indicate that they are positive and support the manuscript's claims.

I encourage the authors to take this as an opportunity to have fleshed out their submission from reviewer suggestions, to integrate their significant new experiments into the manuscript, and submit to the next venue where their work can be fully evaluated.

**Justification For Why Not Higher Score:**

Half of the results in the manuscript are essentially in the author responses. I don't see how one can fairly evaluate this work at present.

**Justification For Why Not Lower Score:**

N/A

---

### Decision · Program_Chairs · 2024-01-16

Reject